# Doping Characteristics and Band Engineering of InSe for Advanced Photodetectors: A DFT Study

**DOI:** 10.3390/nano15100720

**Published:** 2025-05-10

**Authors:** Wenkai Zhang, Yafei Ning, Hu Li, Chaoqian Xu, Yong Wang, Yuhan Xia

**Affiliations:** 1School of Integrated Circuits, Shandong University, Jinan 250101, China; wkzhang@mail.sdu.edu.cn (W.Z.); hu.li@sdu.edu.cn (H.L.); 2School of Geodesy and Geomatics, Wuhan University, Wuhan 250101, China; chqxu@sgg.whu.edu.cn; 3School of Space Science and Technology, Institute of Space Sciences, Shandong University, Weihai 264209, China; wangyong180@163.com; 4School of Electrical and Electronic Engineering, Nanyang Technological University, Singapore 699010, Singapore; yxia011@e.ntu.edu.sg

**Keywords:** DFT study, Ag/Bi doping, InSe, bandgap modulation, electronic structure

## Abstract

Two-dimensional materials have emerged as core components for next-generation optoelectronic devices due to their quantum confinement effects and tunable electronic properties. Indium selenide (InSe) demonstrates breakthrough photoelectric performance, with its remarkable light-responsive characteristics spanning from visible to near-infrared regions, offering application potential in high-speed imaging, optical communication, and biosensing. This study investigates the doping characteristics of InSe using first-principles calculations, focusing on the doping and adsorption behaviors of Argentum (Ag) and Bismuth (Bi) atoms in InSe and their effects on its electronic structure. The research reveals that Ag atoms preferentially adsorb at interlayer vacancies with a binding energy of −2.19 eV, forming polar covalent bonds. This reduces the band gap from the intrinsic 1.51 eV to 0.29–1.16 eV and induces an indirect-to-direct band gap transition. Bi atoms doped at the center of three Se atoms exhibit a binding energy of −2.06 eV, narrowing the band gap to 0.19 eV through strong ionic bonding, while inducing metallic transition at inter-In sites. The introduced intermediate energy levels significantly reduce electron transition barriers (by up to 60%) and enhance carrier separation efficiency. This study links doping sites, electronic structures, and photoelectric properties through computational simulations, offering a theoretical framework for designing high-performance InSe-based photodetectors. It opens new avenues for narrow-bandgap near-infrared detection and carrier transport optimization.

## 1. Introduction

Owing to quantum confinement effects along their atomic-layer thickness, two-dimensional (2D) materials exhibit properties distinct from their three-dimensional counterparts, garnering widespread attention from both academia and industry [1,2,3,4,5,6]. With the rise of research on 2D materials, mechanical exfoliation has been applied to prepare InSe [7,8]. This method enables the isolation of InSe down to atomic thickness, revealing its unique physical characteristics [9,10]. InSe features a graphene-like layered structure composed of vertically stacked Se–In–In–Se layers bound by weak van der Waals forces [11,12]. Depending on stacking sequences, it exhibits β, γ, and ε phases, with a monolayer thickness of approximately 0.85 nm [13,14,15,16,17]. InSe possesses a low electron effective mass, strong quantum confinement effects, and a high room-temperature electron mobility (~10^3^ cm^2^·V^−1^·s^−1^) [18,19,20]. Its bandgap varies widely with layer count (1.26–2.11 eV), making it a compressible 2D semiconductor with high electron mobility and broad visible-to-near-infrared photoluminescence [9,14,15,16,17,18,19,20,21]. Under external stress, InSe undergoes lattice distortion, enabling strain engineering to modulate key parameters such as carrier mobility, photoluminescence quantum yield, and Curie temperature, potentially inducing novel phenomena like topological phase transitions and superconductivity [17,22,23,24]. Notably, InSe’s remarkable photoresponse spanning visible to near-infrared wavelengths demonstrates immense potential in high-speed imaging, optical communication, and biosensing applications [25,26,27].

Recent experimental studies have preliminarily validated InSe’s superior optoelectronic performance [20,28,29,30]. A Peking University team developed high-transconductance field-effect transistors using InSe channels that achieved an 83% room-temperature ballistic ratio at 0.5V operation voltage, surpassing silicon-based device limits [31]. Mechanistic studies reveal that InSe’s interlayer exciton lifetime reaches nanosecond scales, with photogenerated carrier recombination efficiency exceeding transition metal dichalcogenides like MoS_2_ by two orders of magnitude [13,32,33,34,35,36]. However, intrinsic InSe still faces critical challenges in photodetection applications [37,38]. First, its indirect bandgap nature results in a lower optical absorption coefficient compared to direct bandgap materials, limiting its low-light detection capability [33,38,39,40]. Second, the fixed bandgap width struggles to meet the spectral requirements for multi-band photodetection [15,41]. To address these bottlenecks, doping engineering has proven to be an effective approach for tuning the optoelectronic properties of 2D materials—introducing foreign atoms can precisely modulate the material’s Fermi level position, carrier concentration, and band structure [42,43]. For instance, Ag-doped MoS_2_ extends photoresponse wavelengths to 1550 nm, while Bi-doped black phosphorus enhances carrier mobility by 300% [36,44]. These successful examples demonstrate that exploring doping effects in InSe holds significant scientific value for overcoming its optoelectronic performance limits [30,45,46,47]. This study investigates the regulatory mechanisms of Ag/Bi doping on InSe’s optoelectronic properties. First-principles calculations reveal that Ag atoms preferentially occupy interlayer vacancies, forming polarity covalent bonds that reduce the bandgap from 1.51 eV to 0.29–1.16 eV while inducing indirect-to-direct bandgap transitions, enabling spectral range expansion. Bi doping at Se-centered sites (inter-Se) narrows the bandgap to 0.19 eV through ionic bonding, inducing metallic transition at specific sites (inter-In). Remarkably, doping-induced intermediate energy levels reduce electron transition barriers by 60%, significantly enhancing carrier separation efficiency. This fundamental exploration establishes critical theoretical frameworks and strategic pathways for advancing InSe-based photodetectors [47].

## 2. Materials and Methods

### 2.1. Structural Modeling and Calculations

InSe is a two-dimensional periodic structure [17,48]. To perform in-plane doping and adsorption calculations on a monolayer InSe, the structural model was constructed by first cleaving the InSe crystal along the (001) direction [10,49]. A 20 Å vacuum layer was added to eliminate interlayer interactions for studying surface adsorption properties. The InSe plane was oriented in the xy-direction, with the z-axis as the normal direction. To enhance computational efficiency, we performed geometry optimization on four models (2 × 1 × 1, 2 × 2 × 1, 3 × 2 × 1, and 4 × 4 × 1) and found that the 2 × 2 × 1 supercell can accurately reflect the properties of doped atoms in InSe. In this study, we primarily considered the interactions and effects between nearest neighboring atoms and the doped atoms. In order to maximize computational efficiency and consider the influence of neighboring atoms on adsorbed atoms, we established 12-atom supercell models of pristine monolayer InSe, which includes 6 Se atoms and 6 In atoms. The 12-atom supercell was built by a 2×2×1 pristine monolayer InSe cell as shown in Figure 1. For adsorption models, the initial positions of adsorbed atoms (X = Ag, Bi) were set 2 Å above the adsorption centers along the z-axis. The doping rate of adsorbed atoms in the pristine monolayer InSe is 8.33%. Typically, adsorbed atoms occupy high-symmetry sites, while dopant atoms are located in high-symmetry vacancies within InSe. These configurations generally minimize the system energy. Consequently, eight high-symmetry positions were identified for the InSe model, with their corresponding structures illustrated in Figure 2.

### 2.2. Parameter Settings and Convergence Tests

The calculations were performed using density functional theory (DFT) based on first-principles to investigate doping effects and the adsorption energy of metal atoms on intrinsic InSe. The Cambridge Sequential Total Energy Package (CASTEP) module in the Materials Studio 2020 software was employed for geometric optimization and electronic structure calculations, utilizing plane-wave pseudopotentials within the DFT framework. Due to the significant electron density variations at adsorption sites in the models, the generalized gradient approximation (GGA) with the Perdew–Burke–Ernzerhof (PBE) functional was adopted to optimize adsorption configurations and compute energies for stable structures, ensuring accuracy [32,49]. To balance precision and computational efficiency, the energy convergence criterion was set to 10^−5^ eV, with energy gradients at 10^−3^ eV/nm and atomic displacements at 10^−3^ nm [50]. The self-consistent field (SCF) charge density convergence threshold was set to 10^−6^ eV [10,32]. For high-precision calculations on small supercells (2 × 2 × 1), norm-conserving pseudopotentials enable accurate determination of charge transfer, band structures, and electronic density of states, while demonstrating excellent performance in simulating systems with strongly correlated effects. The pseudopotential is set to norm-conserving for geometric optimization [49,50,51,52,53]. InSe exhibits a layered hexagonal symmetry with the space group P63/mmc [54]. In doping simulations, fixing vertical parameters maintains structural stability [55,56]. This strategy preserves structural integrity while significantly reducing computational overhead. Guided by the symmetry requirements of monolayer InSe, the in-plane lattice parameters a and b were allowed to relax during structural optimization, whereas the vertical lattice constant c and angles α, β remained constrained [56,57,58]. All atoms were freely relaxed during optimization, with spin polarization considered.

To ensure computational reliability and minimized runtime, convergence tests were conducted for key parameters, including cutoff energy and k-point sampling. Higher k-point density and cutoff energy improve total energy accuracy but increase computational load. For the Ag adsorption case on pristine InSe, convergence tests (Figure 3) revealed that a k-point mesh of 6 × 6 × 1 and a cutoff energy of 800 eV achieved sufficient energy convergence. Further refinement of these parameters yielded negligible energy changes, confirming their adequacy for balancing accuracy and efficiency. These settings were thus adopted for subsequent calculations.

### 2.3. Binding Energy and Electron Differential Density

Binding energy is the energy required to bring two or more particles from a separated state to a bound state, or equivalently, the energy needed to disrupt such a bond [32]. It reflects the strength and stability of molecular or atomic interactions [10,31]. Typically, binding energy is negative, representing the energy released during the bonding process. A smaller binding energy indicates a more stable bonded configuration. After model construction and optimization, the steady-state energy and equilibrium adsorption distance were calculated to identify the most stable configuration (the state with the lowest energy). Subsequently, the binding energy was determined by computing the steady-state energies of the dopant atoms and InSe within the unit cell.

For the case of atoms adsorbed on the surface of InSe, the adsorption energy corresponds to the binding energy. The adsorption energy is calculated as follows:$$E_{\mathrm{ads}}=E_{\mathrm{InSe}+X}-E_{InSe}-E_{X}$$
where E_InSe_ is the energy of pristine InSe, E_X_ is the energy of the isolated adsorbate atom, and E_InSe+X_ is the total energy of the adsorption system. Typically, adsorption with energies above 100 kJ/mol (approximately 1.037 eV) is considered as chemical adsorption [32,59]. Chemical adsorption typically involves electron exchange or transfer between the adsorbate and substrate [60,61]. Therefore, if adsorbed atoms exhibit electronic interactions with InSe and the adsorption energy is less than −1.037 eV, we classify the adsorption as chemisorption.

For the case of atoms doped into vacancies within InSe, the binding energy calculation is analogous to adsorption energy. The formula for binding energy is as such:$$E_{\mathrm{inter}}=E_{\mathrm{InSe}+X}-E_{InSe}-E_{X}$$
where E_InSe_ is the energy of pristine monolayer InSe, E_X_ is the energy of the isolated dopant atom, E_InSe+X_ is the total energy of the doped system, and E_inter_ represents the binding energy for interstitial doping. For substitutional doping (atoms replacing host atoms in InSe), the binding energy is calculated as follows:$$E_{\mathrm{sub}}=E_{\mathrm{InSe}+X}-\left(E_{InSe}+\mu_{X}-\mu_{Y}\right)$$
where μ_Y_ and μ_X_ are the chemical potentials of the substituted host atom (Y = Se, In) and the dopant atom (X = Ag, Bi) at T = 0, respectively. E_sub_ denotes the binding energy for substitutional doping.

Electron differential density is a key tool for analyzing changes in electronic structure, primarily used to study chemical bond formation, charge transfer, or electron redistribution between a system and its reference state [62]. Electron differential density (Δρ) is defined as the difference between the actual charge density of the system and the charge density of the reference state [63].$$\Delta \rho =\rho_{\mathrm{total}}-\sum \rho_{\mathrm{atom}\;i}$$
where ρ_total_ is the total charge density of the optimized system and ρ_atom i_ denotes the charge density of an isolated atom. For regions where Δρ > 0 (typically represented in red), electrons accumulate in these areas, which generally correspond to the electron-sharing regions of chemical bonds (covalent or coordination bonds) or the acceptor regions of charge transfer. For regions where Δρ < 0 (typically represented in blue), electrons are depleted in these areas, which usually represent donor regions of charge transfer or unoccupied regions of atomic orbitals prior to bonding.

## 3. Results and Discussion

### 3.1. Doping Structures of InSe

To investigate the stability of various doping configurations in InSe and analyze the doping processes, we calculated the binding energies of different Ag-doped configurations. As shown in Figure 4, the Ag atom adsorbed above the central vacancy of InSe (ads-H) exhibits the lowest binding energy of −2.19 eV. The configurations where Ag is doped into the central vacancy (inter-H) or adsorbed at the three-In-atom central vacancy (inter-In) both show slightly higher binding energies of −1.80 eV. Notably, substitutional doping at Se sites (sub-Se) and In sites (sub-In) display significantly higher binding energies of 1.97 eV and 0.05 eV, respectively, indicating much lower stability compared to other configurations.

The positive binding energies of substitutional doping (sub-Se, sub-In) suggest these configurations are energetically unfavorable. Structural optimization reveals that when Ag is doped between three In atoms (inter-In), the atom shifts downward toward the central vacancy, implying the inter-In configuration cannot stably exist. This indicates that the inter-In and inter-H configurations essentially represent the same state. Therefore, the most stable configurations are Ag adsorption above the central vacancy (ads-H) and doping into the central vacancy (inter-H). The doping/adsorption priority sequence in InSe is determined as follows: Ag preferentially adsorbs above the central vacancy (ads-H), followed by doping into the central vacancy (inter-H). Subsequent options include adsorption above Se atoms (ads-Se), In atoms (ads-In), and doping into three-Se-atom vacancies (inter-Se), with substitutional doping (sub-Se, sub-In) being the least favorable. This hierarchy reflects the relative stability and feasibility of different Ag incorporation modes in InSe.

Furthermore, the adsorption energies of Ag atoms in all three adsorption configurations on InSe are below −1.037 eV, and the electron density difference maps demonstrate significant electronic interactions between the Ag atoms and the InSe substrate. These observations confirm that the adsorption of Ag atoms on InSe corresponds to chemisorption. Under the Pauling scale, the electronegativities of Ag, Se, and In atoms are 1.93, 2.55, and 1.78, respectively [64]. Generally, when the electronegativity difference (Δχ) exceeds 1.7, the bond is considered ionic, while Δχ < 1.7 indicates a covalent bond [65]. A larger Δχ corresponds to stronger polarity and partial ionic character in covalent bonds. The electronegativity of Ag is higher than In but lower than Se, with Δχ < 1.7, leading to covalent bonding. Although the electronegativity difference between Se and Ag is significant, the smaller atomic radius of Ag compared to Se induces ionic polarization, weakening electrostatic ionic interactions and enhancing covalent character.

As shown in Figure 5, electron differential density analysis of various configurations reveals that the bonds formed between Ag and Se/In exhibit low polarity, with electron density concentrated between the atoms rather than skewed toward one side. Notably, the interaction regions between Ag and Se atoms exhibit higher electron density, indicating stronger Ag–Se interactions and the formation of stable covalent bonds with weak polarity. In substitutional doping, when Ag replaces In, electron density concentrates between Ag–Se bonds, while the electron density around Se atoms decreases due to charge redistribution toward the Ag–Se interaction region. Conversely, when Ag replaces Se, the electron density accumulates between Ag–In bonds, with minimal changes in electron density around neighboring In atoms. These observations demonstrate that Ag atoms preferentially occupy central vacancies in InSe, forming stable polar covalent bonds with both In and Se atoms. This bonding configuration balances charge distribution and achieves structural stability through symmetrical electron sharing between Ag and its neighboring atoms.

As shown in Figure 6, the Bi atom doped into the central vacancy of InSe (inter-H) exhibits the lowest binding energy of −2.06 eV. The binding energies sequentially increase for configurations including Bi adsorbed at the three-Se-atom central vacancy (inter-Se), Bi adsorbed above the central vacancy (ads-H), Bi adsorbed above Se atoms (ads-Se), Bi adsorbed at the three-In-atom central vacancy (inter-In), and Bi adsorbed above In atoms (ads-In). Notably, substitutional doping of Bi at Se sites (sub-Se) shows a positive binding energy of 0.96 eV, indicating structural instability and low feasibility. The most stable configurations are Bi doping into the central vacancy (inter-H) and three-Se-atom vacancy (inter-Se), followed by adsorption above the central vacancy (ads-H) and Se atoms (ads-Se). Less favorable configurations include adsorption above In atoms (ads-In) and doping into the three-In-atom vacancy (inter-In), with substitutional doping (sub-Se, sub-In) being the least thermodynamically accessible. Bi atoms preferentially occupy the central vacancy (inter-H) or the center of three Se atoms (inter-Se), followed by adsorption above the InSe center (ads-H) or Se atoms (ads-Se). Adsorption above In atoms (ads-In) and doping between three In atoms (inter-In) are less favorable, with substitutional doping (sub-Se, sub-In) being the least stable.

Similar to the Ag atoms, all three adsorption configurations of Bi atoms on InSe exhibit adsorption energies below −1.037 eV, and significant electronic interactions between the Bi atoms and the InSe substrate are observed in electron density difference maps. These results conclusively classify the adsorption of Bi atoms on InSe as chemisorption. Under the Pauling scale, the electronegativity of Bi is 2.02 [64]. The electronegativity of Bi is relatively close to that of In, resulting in covalent bonds with weak polarity. In contrast, the electronegativity difference between Bi and Se is larger, leading to stronger polarity in the covalent bonds. Additionally, the similar atomic radii of Bi and Se result in weaker ionic polarization, which enhances the covalent bond polarity while still retaining partial ionicity. As shown in Figure 7, Bi doping introduces a certain degree of symmetry breaking in InSe. This phenomenon likely arises from the heavy-metal nature of Bi, which exhibits strong spin–orbit coupling (SOC) [66]. The SOC effect further modifies the electronic density distribution in the Bi-doped system. The electron differential density analysis of various configurations reveals that the bonds formed between Bi and In atoms exhibit weak polarity with minimal electron exchange, whereas interactions between Bi and Se atoms result in highly polarized bonds.

By comparing the geometrically relaxed structures (Figure 8) it can be observed that when Bi atoms are doped around three In atoms (inter-In), the Bi atoms shift downward to the central vacancy of InSe, resembling the structure doped at the central vacancy (inter-H). Therefore, we conclude that the two configurations geometrically converge during relaxation. Bi doping at the center of three In atoms is unstable, and the Bi atom tends to migrate toward the center of InSe to achieve stabilization. When Bi substitutes Se, the electron density around Bi atoms increases, while neighboring In atoms experience a slight electron density reduction. Conversely, when Bi replaces In, electron density around Bi atoms decreases significantly, with charge redistribution toward Bi-In and Bi-Se interaction regions, accompanied by increased electron density around In atoms.

These observations indicate that Bi preferentially interacts electronically with Se atoms. Substitutional doping configurations demonstrate higher stability compared to adsorption configurations, as they enable stronger charge redistribution and more favorable bonding environments. This explains why Bi atoms exhibit greater thermodynamic stability when incorporated into InSe lattices through doping rather than surface adsorption.

Through the analysis of binding energies and electron density differences discussed above, we can determine the relative stability of different doping sites for various atoms. As evidenced by existing studies, experimental strategies such as modulating reaction conditions or utilizing surfactants can guide the adsorption or doping of atoms at specific sites [67,68]. Although substitutional doping in pristine InSe exhibits higher binding energies and is therefore less favorable compared to other configurations, the presence of In or Se vacancy defects during experimental synthesis significantly reduces the energy barrier for substitutional doping [69,70]. Consequently, introducing Ag or Bi atoms into defective InSe enables the formation of substitutional doping structures.

### 3.2. Band Structure Analysis

To further analyze the effects of dopant atoms on InSe, the band structures of various doped configurations were calculated. As shown in Figure 9, intrinsic InSe exhibits typical semiconductor characteristics with the Fermi level located at the mid-gap and a bandgap of approximately 1.51 eV. Due to the neglect of long-range exact exchange interactions in the GGA-PBE functional, the bandgap values calculated using GGA-PBE are inherently underestimated. To ensure the reliability of our calculations, we compared our results with those reported in existing studies. As shown in Figure 7, in the case of pristine InSe, the material exhibits clear semiconducting behavior, with the Fermi level positioned at the mid-gap and a bandgap of approximately 1.51 eV. This value closely matches the range reported in prior literature (1.26–2.11 eV), validating the consistency of our computational approach [9,17,18,19,20,21].

Figure 10 presents the band structures for Ag doping/adsorption configurations, with corresponding bandgap values quantitatively tabulated in Table 1. When Ag atoms are adsorbed above the central vacancy or Se atoms in InSe, a new impurity energy level is introduced near mid-gap compared to intrinsic InSe. The energy level distribution becomes more concentrated, narrowing the bandgap by 0.54 eV and 0.52 eV, respectively. This enables electrons to transition more easily from the valence band maximum (VBM) to the conduction band minimum (CBM) through the new mid-gap levels introduced by Ag.

For Ag adsorption above In atoms, the impurity level appears close to the CBM, effectively lowering the CBM, while reducing its width by 0.5 eV. The band structures of Ag-doped configurations between three In atoms and at the central vacancy are nearly identical, confirming that Ag atoms adsorbed between three In atoms are unstable and tend to migrate to the central vacancy. In this case, both the conduction and valence bands show significantly increased state density, with the bandgap reduced to 1.16 eV.

When Ag atoms are adsorbed between three Se atoms, partial valence band energy levels rise above the Fermi level and overlap with the conduction band, resulting in metallic-like properties. The bandgap becomes direct and dramatically shrinks to 0.29 eV. For substitutional doping, Ag replacing Se introduces impurity levels near the CBM, further narrowing the bandgap. Ag substituting in introduces multiple mid-gap impurity levels, facilitating electron transitions from the VBM to the CBM through these intermediate states.

Figure 11 presents the band structures for Bi doping/adsorption configurations, with corresponding bandgap values quantitatively tabulated in Table 2. After doping or adsorption, the energy level distribution becomes more concentrated. When Bi atoms are adsorbed above Se atoms, the impurity energy level approaches the CBM, effectively lowering the CBM and reducing the band gap by 0.02 eV, transforming it into a direct bandgap. For Bi atoms adsorbed above the central vacancy in InSe, impurity levels introduced near the mid-gap facilitate electron transitions, resulting in a narrowed direct bandgap of 0.73 eV. When Bi atoms are adsorbed above In atoms, impurity levels near the VBM reduce the band gap by 0.59 eV, forming a direct bandgap. Substitutional doping of Bi at the central vacancy introduces impurity levels near the CBM, decreasing the bandgap to 1.05 eV. Doping Bi at three Se sites introduces mid-gap impurity levels, significantly reducing the bandgap to 0.49 eV with a direct transition. When Bi atoms are doped between three In atoms, the VBM rises above the Fermi level while the CBM drops below it, endowing InSe with metallic characteristics, enhanced conductivity, and an ultranarrow band gap of 0.19 eV. Substitutional doping of Bi at Se sites introduces impurity levels near the VBM, reducing the band gap by 0.66 eV to form a direct gap. Similarly, Bi substitution at In sites introduces CBM-proximate impurity levels, decreasing the band gap by 0.53 eV. Furthermore, significant changes in the band structure are observed in Bi-doped or adsorbed configurations, which predominantly favor the formation of indirect band gaps. This behavior is likely attributed to the SOC of Bi atoms, where relativistic effects induce band splitting by coupling the electron’s spin and orbital angular momentum [66]. Such symmetry-breaking effects, combined with Bi’s SOC, could theoretically induce or amplify second-order nonlinear optical responses.

Additionally, we further elucidate the variation in intrinsic carrier concentration by analyzing the changes in the density of states (DOS) before and after doping (Figure 12). By comparing the DOS of InSe with and without Ag doping, it is observed that when Ag is adsorbed above the In atom (Ag:ads-In) or adsorbed at the center of three Se atoms (Ag:inter-Se), the Fermi level shifts toward the conduction band. This introduces impurity energy levels that facilitate electron transitions from the VBM to the CBM. Similarly, when Bi atoms are adsorbed above the Se atom (Bi:ads-Se), the Fermi level shifts toward the conduction band, accompanied by a higher DOS near the CBM. The enhanced DOS near the CBM likely leads to an increase in electron concentration, improving the conductivity of InSe and enabling carriers to participate more effectively in charge transport. These results demonstrate that Ag/Bi doping/adsorption in InSe reduces the bandgap and introduces impurity levels that promote electron transitions, potentially increasing carrier concentration. We thus infer that InSe’s conductivity improves with greater carrier involvement in charge transport. Additionally, bandgap engineering via doping can convert indirect-to-direct transitions, enhancing carrier recombination efficiency. By strategically modulating Ag/Bi doping sites, the bandgap width and type can be tailored to optimize light absorption efficiency for specific wavelengths, enabling performance customization in photodetectors. Site-selective doping further allows targeted optimization of photoresponse at desired wavelengths.

## 4. Conclusions

This study systematically investigates the doping characteristics of Ag and Bi atoms in InSe and their regulatory mechanisms on optoelectronic properties through first-principles calculations, revealing the following key conclusions:

(1) Ag atoms preferentially adsorb at the interlayer vacancy of InSe (ads-H, binding energy: −2.19 eV), forming weakly polar covalent bonds with In/Se atoms. In contrast, Bi atoms stably occupy the central site of three Se atoms (inter-Se, binding energy: −2.06 eV), establishing polar covalent bonds with partial ionic character.

(2) Doping significantly modulates the band structure of InSe: Ag doping reduces the intrinsic bandgap from 1.51 eV to 0.29–1.16 eV and induces an indirect-to-direct bandgap transition at the inter-Se site. Bi doping further compresses the bandgap to 0.19 eV via intermediate energy levels and exhibits metallic characteristics at the inter-In site.

(3) Theoretical calculations demonstrate that the intermediate energy levels introduced by doping reduce the electron transition barrier by up to 60% and enhance carrier generation efficiency through increased density of states at the conduction band minimum, providing a novel strategy for optimizing near-infrared photoresponse.

This study elucidates the regulatory mechanisms of Ag/Bi doping on the electronic structure of InSe, providing critical theoretical guidance for the experimental design and performance optimization of high-performance InSe-based electronic devices. To further investigate the doping properties of InSe, future work should integrate experimental validation of structural stability. By modulating reaction conditions or surfactants, dopant atoms could be guided to adsorb or incorporate at specific sites, enabling precise control over the bandgap width and type (direct/indirect) of InSe. Such precise bandgap engineering could achieve orders-of-magnitude enhancements in optical absorption coefficients and spectral customization. These insights not only establish a physical foundation for breakthroughs in InSe-based photodetectors but also propose a universal design paradigm for tailoring optoelectronic properties in two-dimensional materials. While this work relies on computational simulations to analyze doping effects, experimental measurements are required to quantify the full range of bandgap modulation and carrier concentration variations, which will be addressed in subsequent studies.

## Figures and Tables

**Figure 1 nanomaterials-15-00720-f001:**
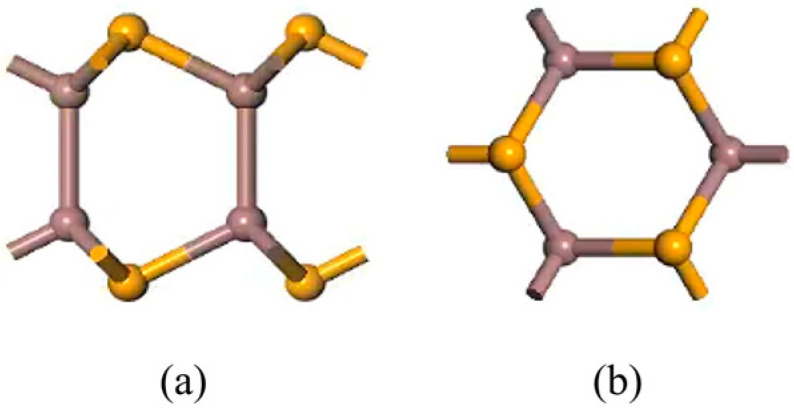
Models of pristine monolayer InSe. (**a**) shows top views of pristine monolayer InSe and (**b**) shows side views of pristine monolayer InSe.

**Figure 2 nanomaterials-15-00720-f002:**
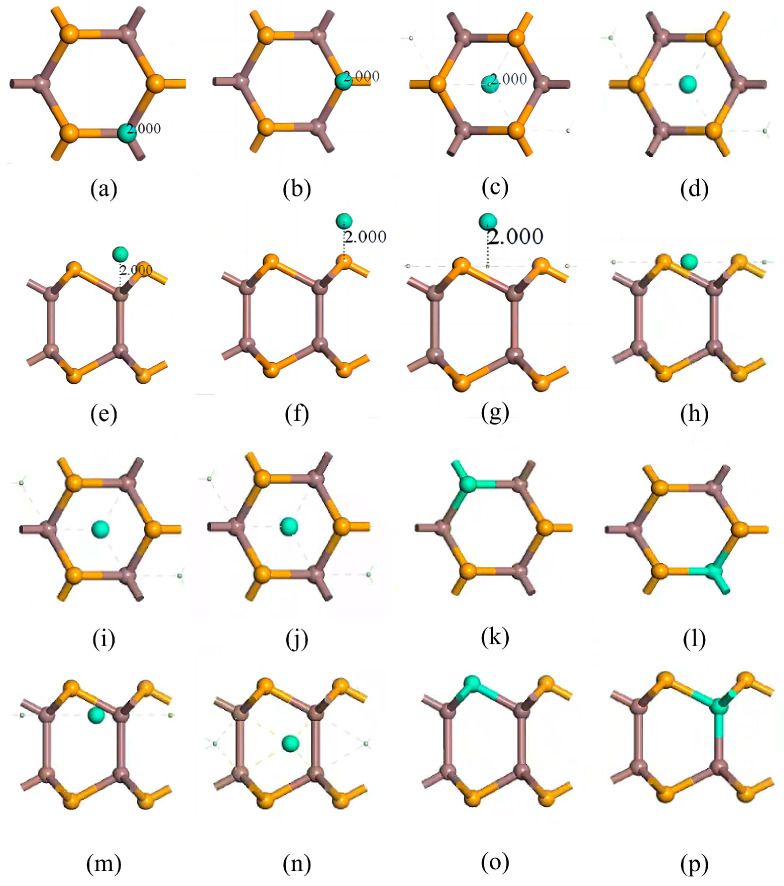
Eight doping/adsorption models of InSe. (**a**,**e**) doping atoms adsorbed above In atoms (ads-In). (**b**,**f**) doping atoms adsorbed above Se atoms (ads-Se). (**c**,**g**) doping atoms adsorbed above the central vacancy (ads-H). (**d**,**h**) doping atoms adsorbed at the center of three Se atoms (inter-Se). (**i**,**m**) doping atoms adsorbed at the center of three In atoms (inter-In). (**j**,**n**) doping atoms incorporated into the central vacancy (inter-H). (**k**,**o**) doping atoms substitutionally doped at Se sites (sub-Se). (**l**,**p**) doping atoms substitutionally doped at In sites (sub-In). (**a**–**d**,**i**–**l**) show top views; (**e**–**h**,**m**–**p**) show side views. Atom color coding: yellow = Se, brown = In, green = doping atoms (X = Ag, Bi).

**Figure 3 nanomaterials-15-00720-f003:**
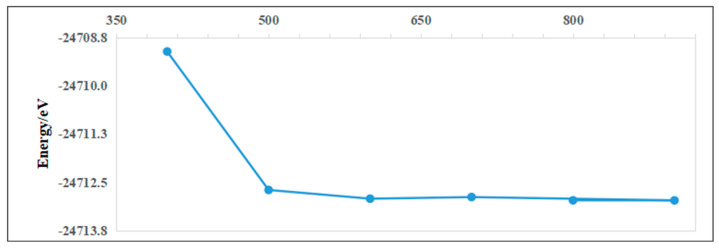
Taking the adsorption case of Ag on pristine InSe as an example. The upper panel shows the cutoff energy convergence test, and the lower panel shows the k-point convergence test.

**Figure 4 nanomaterials-15-00720-f004:**
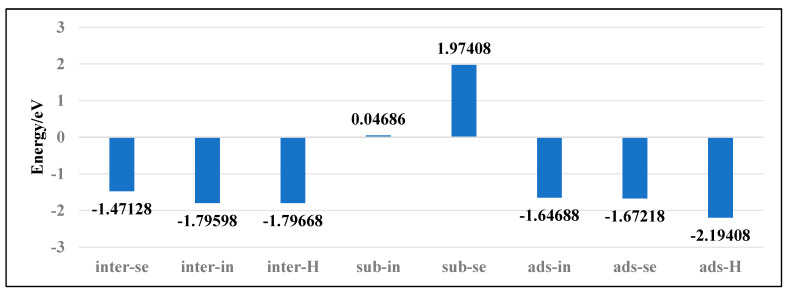
Binding energies of Ag atoms in doping and adsorption configurations in InSe.

**Figure 5 nanomaterials-15-00720-f005:**
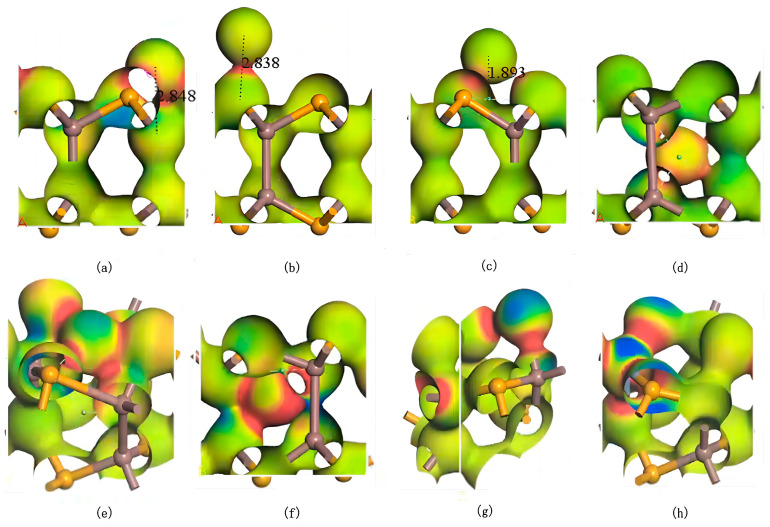
Electron differential density maps of Ag atoms in doping and adsorption configurations in InSe. Regions in red indicate electron accumulation, while blue regions indicate electron depletion. (**a**) Ag atom adsorbed above the In atom in InSe (ads-In). (**b**) Ag atom adsorbed above the Se atom in InSe (ads-Se). (**c**) Ag atom adsorbed above the central vacancy in InSe (ads-H). (**d**) Ag atom incorporated into the central vacancy in InSe (inter-H). (**e**) Ag atom adsorbed at the center of three Se atoms in InSe (inter-Se). (**f**) Ag atom adsorbed at the center of three In atoms in InSe (inter-In). (**g**) Ag atom substitutionally doped at the Se site in InSe (sub-Se). (**h**) Ag atom substitutionally doped at the In site in InSe (sub-In). The numbers in (**a**–**c**) represent the geometrically optimized distances between the adsorbed atoms and their respective adsorption sites.

**Figure 6 nanomaterials-15-00720-f006:**
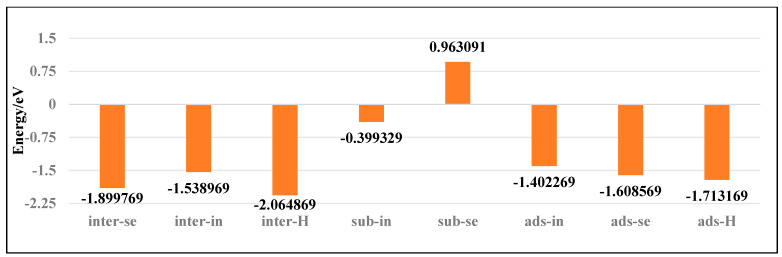
Binding energies of Bismuth (Bi) atoms in doping and adsorption configurations in InSe.

**Figure 7 nanomaterials-15-00720-f007:**
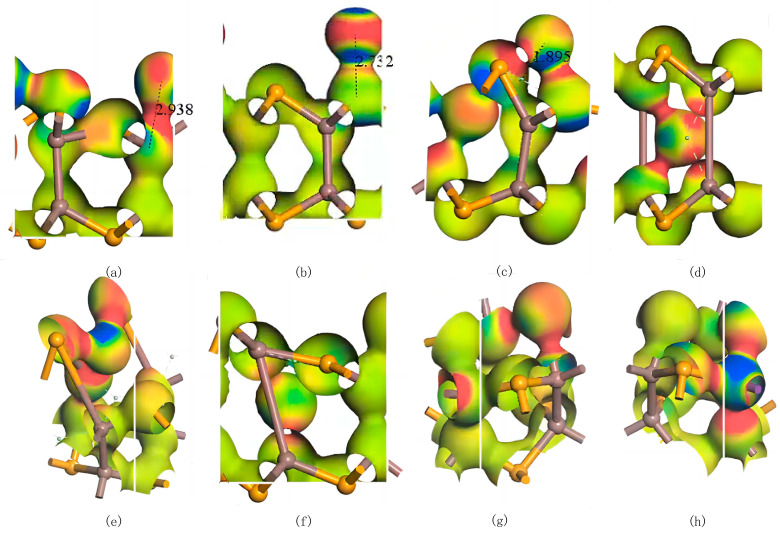
Electron differential density maps of Bismuth (Bi) atoms in doping and adsorption configurations in InSe. Red regions indicate electron accumulation, while blue regions denote electron depletion. (**a**) Bi atom adsorbed above the In atom in InSe (ads-In). (**b**) Bi atom adsorbed above the Se atom in InSe (ads-Se). (**c**) Bi atom adsorbed above the central vacancy in InSe (ads-H). (**d**) Bi atom incorporated into the central vacancy in InSe (inter-H). (**e**) Bi atom adsorbed at the center of three Se atoms in InSe (inter-Se). (**f**) Bi atom adsorbed at the center of three In atoms in InSe (inter-In). (**g**) Bi atom substitutionally doped at the Se site in InSe (sub-Se). (**h**) Bi atom substitutionally doped at the In site in InSe (sub-In). The numbers in (**a**–**c**) represent the geometrically optimized distances between the adsorbed atoms and their respective adsorption sites.

**Figure 8 nanomaterials-15-00720-f008:**
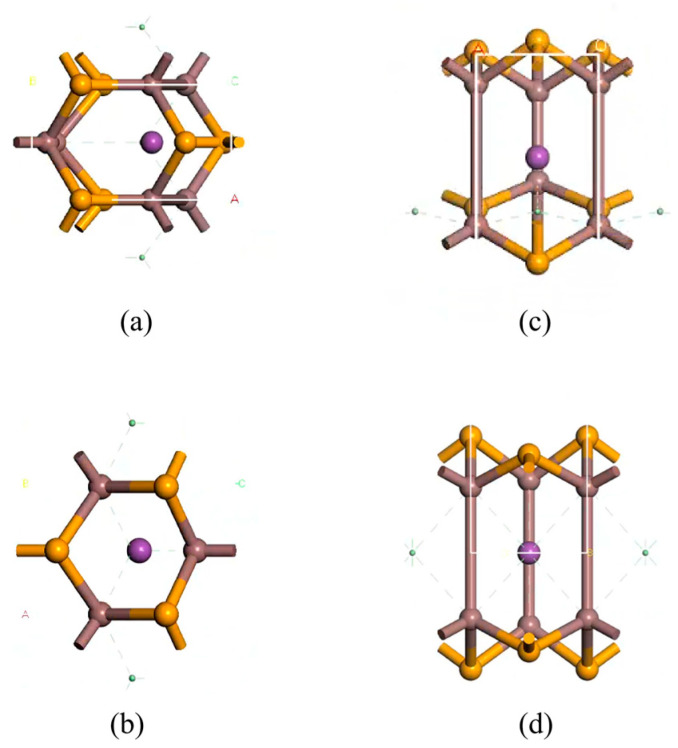
Structural comparison of geometry-optimized Bi-doped configurations: inter-In (Bi atom adsorbed at the center of three In atoms in InSe, top view in (**a**), side view in (**c**), and inter-H Bi atom incorporated into the central vacancy in InSe, top view in (**b**), side view in (**d**).

**Figure 9 nanomaterials-15-00720-f009:**
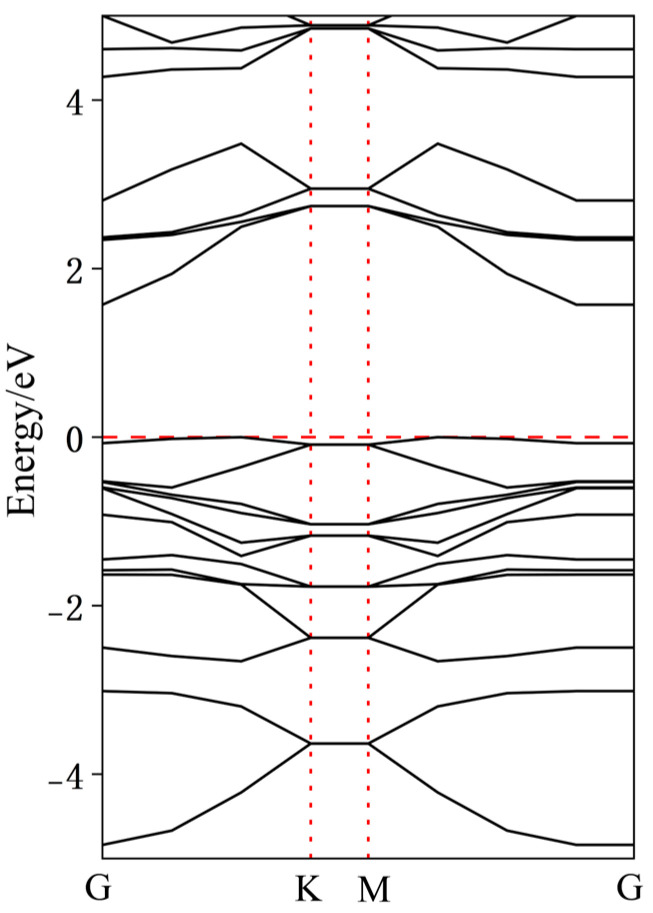
Band structure of pristine monolayer InSe.

**Figure 10 nanomaterials-15-00720-f010:**
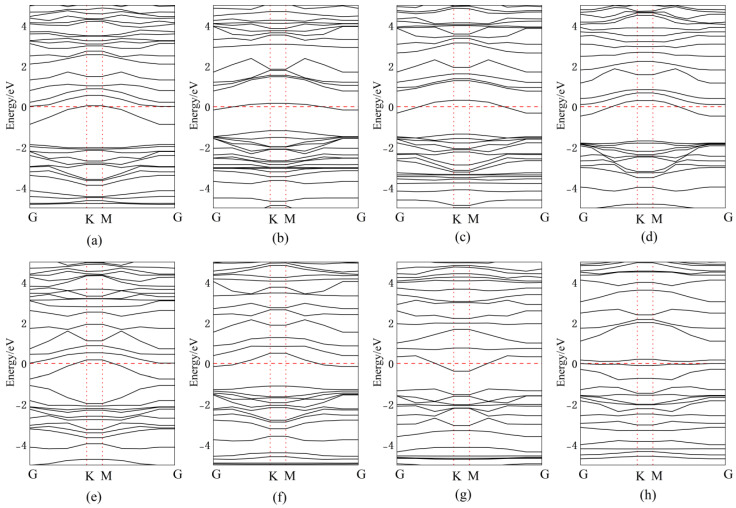
Band structures of Ag atoms in doping and adsorption configurations in InSe. (**a**) Ag atom adsorbed above the In atom in InSe (ads-In). (**b**) Ag atom adsorbed above the Se atom in InSe (ads-Se). (**c**) Ag atom adsorbed above the central vacancy in InSe (ads-H). (**d**) Ag atom incorporated into the central vacancy in InSe (inter-H). (**e**) Ag atom adsorbed at the center of three Se atoms in InSe (inter-Se). (**f**) Ag atom adsorbed at the center of three In atoms in InSe (inter-In). (**g**) Ag atom substitutionally doped at the Se site in InSe (sub-Se). (**h**) Ag atom substitutionally doped at the In site in InSe (sub-In).

**Figure 11 nanomaterials-15-00720-f011:**
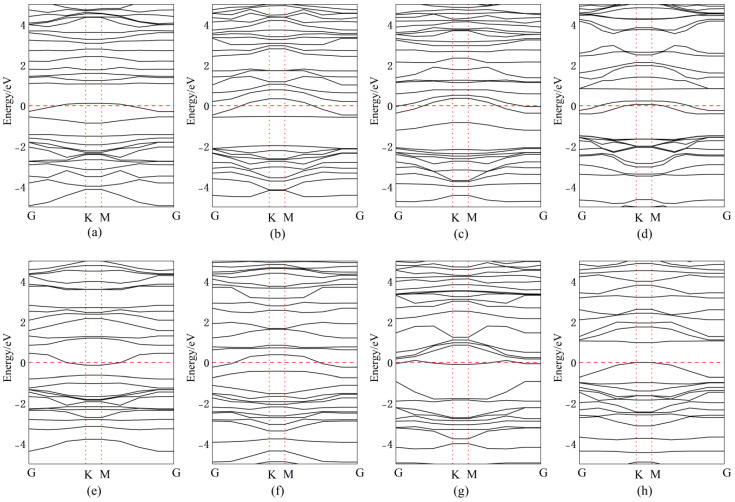
Band structures of Bismuth (Bi) atoms in doping and adsorption configurations in InSe. (**a**) Bi atom adsorbed above the In atom in InSe (ads-In). (**b**) Bi atom adsorbed above the Se atom in InSe (ads-Se). (**c**) Bi atom adsorbed above the central vacancy in InSe (ads-H). (**d**) Bi atom incorporated into the central vacancy in InSe (inter-H). (**e**) Bi atom adsorbed at the center of three Se atoms in InSe (inter-Se). (**f**) Bi atom adsorbed at the center of three In atoms in InSe (inter-In). (**g**) Bi atom substitutionally doped at the Se site in InSe (sub-Se). (**h**) Bi atom substitutionally doped at the In site in InSe (sub-In).

**Figure 12 nanomaterials-15-00720-f012:**
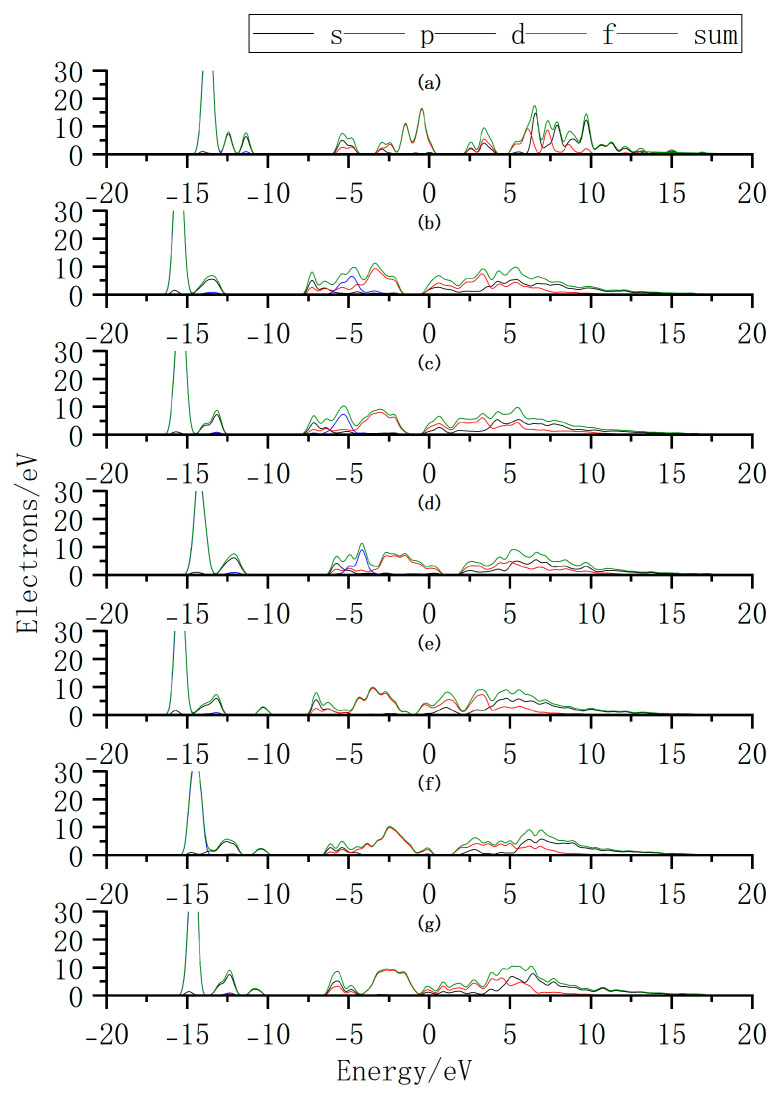
Density of states (DOS) distributions for Ag and Bi atoms doped or adsorbed in InSe. (**a**) Pristine monolayer InSe. (**b**) Ag atom adsorbed above the In atom in InSe (Ag:ads-In). (**c**) Ag atom adsorbed at the center of three Se atoms in InSe (Ag:inter-Se). (**d**) Ag atom substitutionally doped at the In site in InSe (Ag:sub-In). (**e**) Bi atom adsorbed above the Se atom in InSe (Bi:ads-Se). (**f**) Bi atom incorporated into the central vacancy in InSe (Bi:inter-In). (**g**) Bi atom substitutionally doped at the In site in InSe (Bi:sub-Se).

**Table 1 nanomaterials-15-00720-t001:** Bandgap widths and bandgap types of Ag atoms in doping and adsorption configurations in InSe, where ΔE represents the difference between the bandgap width and that of intrinsic InSe (positive values indicate bandgap widening, negative values indicate bandgap narrowing).

Doping Position	E_g_	△E	Bandgap Type
ads-In	1.01 eV	−0.50 eV	Direct bandgap
ads-Se	0.98 eV	−0.52 eV	Indirect bandgap
ads-H	0.96 eV	−0.54 eV	Indirect bandgap
inter-In	0.86 eV	−0.65 eV	Indirect bandgap
inter-Se	0.29 eV	−1.22 eV	Direct bandgap
inter-H	1.16 eV	−0.35 eV	Indirect bandgap
sub-In	0.83 eV	−0.68 eV	Indirect bandgap
sub-Se	0.84 eV	−0.67 eV	Indirect bandgap

**Table 2 nanomaterials-15-00720-t002:** Bandgap widths and bandgap types of Bi Atoms in doping and adsorption configurations in InSe, where ΔE represents the difference between the bandgap width and that of intrinsic InSe (positive values indicate bandgap widening, negative values indicate bandgap narrowing).

Doping Position	E_g_	△E	Bandgap Type
ads-In	0.92 eV	−0.59 eV	Direct bandgap
ads-Se	1.49 eV	−0.02 eV	Direct bandgap
ads-H	0.73 eV	−0.78 eV	Direct bandgap
inter-In	0.19 eV	−1.32 eV	Direct bandgap
inter-Se	0.49 eV	−1.02 eV	Direct bandgap
inter-H	1.05 eV	−0.46 eV	Indirect bandgap
sub-In	0.98 eV	−0.53 eV	Indirect bandgap
sub-Se	0.85 eV	−0.66 eV	Direct bandgap

## Data Availability

The data that support the findings of this study are available from the corresponding author upon reasonable request.

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
