# Peer review of "Doping Characteristics and Band Engineering of InSe for Advanced Photodetectors: A DFT Study"

_nanomaterials, 2025, doi:10.3390/nano15100720_

Round 1
Reviewer 1 Report
Comments and Suggestions for Authors
The authors investigate the doping characteristics of InSe. It is well known that InSe has unique photoelectric properties. The results obtained are of interest for various fields (high-speed imaging, optical communication, biosensing). Thus, the study is relevant. The study used modern theoretical methods. The results can be published. There are comments.
- Figure of the pristine monolayer InSe cell is needed (top view, side view).
- Did the authors use a supercell? If so, what size? Or was doping performed for a single unit cell? What is the impurity concentration?
- “Normconserving pseudopotentials were selected for geometric optimization, and the self-consistent field (SCF) charge density convergence threshold was set to 10⁻⁶ eV[11].” It is necessary to provide a link to the article about the pseudopotential used, and not to the paper of other authors. Or did these authors [11] develop the pseudopotential that was used in this study?
- Are the authors considering chemical adsorption? If the authors are studying physical adsorption, then the Van der Waals interaction must be taken into account.
- Fig. 4 and 6. It is necessary to make high-quality captions for Fig. 4 a b c. For example, in black. The numbers are not visible now.
- “electron differential density”. What formula was used for the calculation?
- “This section is not mandatory but can be added to the manuscript if the discussion is unusually long or complex.” The conclusions can be shortened a little. So far, the authors have presented everything in great detail.
Author Response
Thank you for your valuable suggestions and comments. In response to your feedback, we have carefully revised the manuscript. We have prepared a separate response file addressing your suggestions and comments in detail. You may kindly refer to the attached document for our point-by-point clarifications.

Reviewer 2 Report
Comments and Suggestions for Authors
This manuscript presents a thorough DFT-based investigation of Ag and Bi doping in InSe monolayers for tuning optoelectronic properties, specifically for photodetector applications. The topic is timely and relevant to the 2D materials community. The results are insightful, particularly in showing how doping modulates the bandgap and bonding nature, and could guide experimental work. However, there are several aspects that need clarification, correction, or improvement before the manuscript can be considered for publication.
- Line 22: The phrase "forming weak polar covalent bonds" could be made clearer—does this mean partial ionic/covalent bonding? Consider rewording or elaborating slightly.
- Line 28: “Quantitative correlation model” is mentioned, but no such model or equation is explicitly shown in the manuscript. Please clarify or adjust the wording to better reflect the actual findings.
- The introduction is overly long and somewhat repetitive (e.g., Lines 36–61 vs. 73–91).
- Several citations are grouped awkwardly (e.g., [10,11,24,35–37] in Line 69) and should be distributed more naturally.
- Consider condensing Lines 33–92 into a more focused background that ends with a clear research question.
- Line 122: GGA-PBE is known to underestimate bandgaps. This limitation should be explicitly acknowledged.
- Line 129: The phrase “Lattice parameters c, α, and β were fixed” is ambiguous. Please clarify which lattice parameters were held constant and whether this affects accuracy.
- Did you test HSE06 or any hybrid functionals to better estimate bandgaps?
- Were van der Waals interactions considered, given that InSe is a layered material?
- Line 295–297: You mention that the band structures of inter-In and inter-H are “nearly identical.” Does this suggest geometric convergence during relaxation? Please support this claim with a figure or RMSD metric.
- Line 333: Metallic behavior of inter-In Bi doping is intriguing. Can you elaborate on the partial density of states (PDOS) to explain orbital contributions?
- Line 339: The statement that Ag and Bi “increase intrinsic carrier concentration” is qualitative. Can this be supported with actual density of states or Fermi level shifts?
- What could be the role and impact of structural vacancies?
Author Response

(The authors gave the same response as above.)

Reviewer 3 Report
Comments and Suggestions for Authors
The manuscript titled "Doping Characteristics and Band Engineering of InSe for Advanced Photodetectors: A DFT Study" presents a DFT-based study on Ag and Bi doping in monolayer InSe, exploring multiple configurations and analyzing their influence on structural stability and electronic properties. The manuscript would benefit from improvements in clarity, language, and contextualization within the broader literature, particularly regarding the potential experimental relevance and application of the findings.
A major revision is recommended, primarily to clarify the scope, improve language, and strengthen the positioning of the work within the wider context of InSe-based devices.
1. The use of PBE-GGA is standard in DFT studies, particularly for screening purposes. However, it is well known that GGA functionals tend to underestimate bandgaps in semiconductors, and this can affect conclusions involving absolute values of the electronic structure. I suggest that a brief comment acknowledging the known limitations of PBE-GGA, especially regarding bandgap underestimation, would strengthen the transparency of the study. If feasible, the authors may consider whether selected calculations using hybrid functionals (e.g., HSE06) could further validate the observed trends, particularly for key configurations.
2. While the introduction provides a good overview of experimental interest in InSe, the main text could benefit from a more direct link to specific device-level studies or material behavior that the predictions might impact. Consider adding a short discussion on how Ag or Bi doping might relate to real devices. Additionally, it could be valuable to briefly mention in the introduction that several types of doping (e.g., surface charge transfer, molecular, or electrostatic) have been explored in InSe-based devices, such as in refs.: 10.1002/adfm.201908427, 10.1002/adma.201803690, 10.1002/admi.202201635.
3. The manuscript refers to "enhanced conductivity" and "carrier separation efficiency" due to doping, but it does not include direct calculations of mobility, effective mass, or optical response. The authors should clarify that the claims about device performance are qualitative and based on trends in the electronic band structure (e.g., bandgap narrowing, appearance of impurity states). If desired, a comment on how these predictions could guide future optical or transport studies would add value.
4. While the manuscript analyzes the relative stability of doping configurations based on binding energy, it does not address the kinetic accessibility of those sites (i.e., whether the dopants could realistically reach those positions in practice).
5. Given that bismuth is a heavy element with strong spin–orbit coupling, some Bi-doped configurations may lead to symmetry breaking or enhanced second-order nonlinear optical response (10.1016/j.omx.2023.100255). The authors may consider adding a comment on whether such phenomena could emerge in Bi-doped InSe, particularly in substitutional or interstitial configurations that break symmetry.
6. There are several grammatical and stylistic issues that should be addressed, particularly in the conclusions section. For example:
- "we systematically the investigated"; "we provides"; "modulation electronic"
A full language edit is recommended. Additionally, the conclusion could be rewritten to better summarize the main insights and clearly define what is qualitative (trend-level) vs. what might require experimental confirmation.
Other Minor Suggestions
- Check spacing after punctuation.
- Include a summary table or figure comparing formation energy and bandgap for all configurations.
- Define all abbreviations at first appearance.
There are several grammatical and stylistic issues that should be addressed, particularly in the conclusions section. For example:
- "we systematically the investigated"; "we provides"; "modulation electronic"
A full language edit is recommended. Additionally, the conclusion could be rewritten to better summarize the main insights and clearly define what is qualitative (trend-level) vs. what might require experimental confirmation.
Author Response

(The authors gave the same response as above.)

Round 2
Reviewer 1 Report
Comments and Suggestions for Authors
The authors answered all my questions in great detail. The answers are very well written and formatted. I recommend accepting the paper for publication.
Reviewer 2 Report
Comments and Suggestions for Authors
After successful revision this manuscript can be recommended for publication.
Reviewer 3 Report
Comments and Suggestions for Authors
The point-by-point replies address all reviewer concerns in a convincing way. From a scientific standpoint the manuscript is now suitable for publication. However, several sentences are still overly long and a few “false friends”/literal translations appear. A final copy-edit will improve flow.
Comments on the Quality of English LanguageOverall the English is solid, but several sentences are still overly long and a few “false friends”/literal translations appear. A final copy-edit will improve flow.